# Detection of Broken Rotor Bars in Cage Induction Motors Using Machine Learning Methods

**DOI:** 10.3390/s23229079

**Published:** 2023-11-09

**Authors:** Lloyd Prosper Chisedzi, Mbika Muteba

**Affiliations:** Department of Electrical and Electronic Engineering Technology, University of Johannesburg, Auckland Park, Johannesburg 2006, South Africa; lloydprosperchisedzi@gmail.com

**Keywords:** artificial neural network, broken rotor bar fault detection, current signature, decision tree classifier, deep learning, squirrel cage induction motor

## Abstract

In this paper, the performance of machine learning methods for squirrel cage induction motor broken rotor bar (BRB) fault detection is evaluated. Decision tree classification (DTC), artificial neural network (ANN), and deep learning (DL) methods are developed, applied, and studied to compare their performance in detecting broken rotor bar faults in squirrel cage induction motors. The training data were collected through experimental measurements. The BRB fault features were extracted from measured line-current signatures through a transformation from the time domain to the frequency domain using discrete Fourier Transform (DFT) of the frequency spectrum of the current signal. Eighty percent of the data were used for training the models, and twenty percent were used for testing. A confusion matrix was used to validate the models’ performance using accuracy, precision, recall, and f1-scores. The results evidence that the DTC is less load-dependent, and it has better accuracy and precision for both unloaded and loaded squirrel cage induction motors when compared with the DL and ANN methods. The DTC method achieved higher accuracy in the detection of the magnitudes of the twice-frequency sideband components induced in stator currents by BRB faults when compared with the DL and ANN methods. Although the detection accuracy and precision are higher for the loaded motor than the unloaded motor, the DTC method managed to also exhibit a high accuracy for the unloaded current when compared with the DL and ANN methods. The DTC is, therefore, a suitable candidate to detect broken rotor bar faults on trained data for lightly or thoroughly loaded squirrel cage induction motors using the characteristics of the measured line-current signature.

## 1. Introduction

Most of the electromechanical converters in modern industry are squirrel cage induction motors (IMs), constituting 90% of all electrical motors [1]. Squirrel cage IMs are robust, and they can be used in hazardous environments, such as in oil refineries, blowing operations, etc. A broken rotor bar is one of the common faults in squirrel cage IMs. The BRB fault may be caused by several factors, such as thermal stress, mechanical stress, environmental stress, residual stress, or dynamical stress [2,3]. The approximation rate of fault occurrence is higher for motors rated above 4 kV; about 5% of the fault occurrence rate is due to broken rotor bars in squirrel cage induction motors [3,4]. About 10% of faults that occur in three-phase squirrel cage IMs are due to broken rotor bars [5]. When broken rotor bar faults occur in squirrel cage IMs and go undetected, the current passing through the broken rotor bar will be distributed to the neighboring bars, causing an increase in currents in the healthy bars. This will result in electric arcs being created and a rise in temperature, which may lead to the melting of these rotor bars, thereby further damaging the rotor [6]. Therefore, there will be an adverse effect on the production line, resulting in economic losses, that is why it is so important to continuously monitor the motor’s condition for BRB faults to avoid further damage to the rotor. It is subsequently indispensable for engineers and scholars to keep researching broken rotor bar fault detection methods.

Different faults, such as broken bars, bearing faults, and winding faults, have been studied before, and several fault detection methods, such as support vector machines, neural networks, and decision tree algorithms, have been developed [7,8]. In previous studies, it has been proven that machine learning algorithms such as DTC, ANN, and DL can be used for the detection of different faults. In [8], an envelope analysis of vibration signals using a decision tree algorithm was conducted. The model was built and trained with vibration spectrum data. The authors in [9] compared the deep learning algorithm and decision tree algorithm via image classification for five different vineyards, with images obtained using a drone, and found that the decision tree algorithm outperformed the deep learning algorithm. In [10], an unbalanced voltage fault detection method using a neural network for a three-phase traction motor was presented.

Motor current signature analysis (MCSA) is the most popular method that is used to detect BRB faults in squirrel cage induction motors. Success in the application of MCSA to detect BRB faults in squirrel cage IMs has been reported in [11,12]. The method performs frequency spectrum analysis of the stator currents to determine the electromechanical conditions of the squirrel-cage IM [13]. When a BRB fault occurs, magnetic and geometric imbalances that induce sidebands are produced in the stator current spectrum. This makes it possible to use the analysis of sideband frequencies and their amplitudes for efficiently training machine learning models [11]. There are advantages of using MCSA, including the simplicity of current measuring and sensor installation and the non-requirement of motor parameters [14,15]. Although machine learning methods were used with success in all previous studies, there are still no studies that compared DTC, ANN, and DL for broken rotor bar fault detection using measured line-current signatures through transformation from the time domain to the frequency domain obtained via discrete Fourier Transform (DFT) of the frequency spectrum of the current signal. Hence, this paper extends the literature on MCSA by comparing the performance of three machine learning methods, including DTC, ANN, and DL. These three machine learning models are developed, and their performance is compared to assess their accuracy and precision in the detection of broken rotor bar faults in squirrel cage IMs from measured line-current signatures. It is worth noting that the use of mechanical parameters, such as speed, torque, or vibration, to assess the rotor bar condition of squirrel cage induction motors is not employed in this paper. The organization of this article is as follows: Section 2 presents an overview of the machine learning method. In Section 3, the approach used in this study is presented. The experimental setup, measurements, and broken rotor bar fault extraction are reported in Section 4. The broken rotor bar fault detection using machine learning methods is presented in Section 5, while Section 6 presents a discussion of the results, and Section 7 gives a summary of the key findings.

## 2. Overview of Machine Learning

Machine learning is the science of enabling a computer to learn how to solve a task without being explicitly programmed. Machine learning methods can learn the patterns of non-separable data from training data and predict or detect them at a high accuracy [15]. Different techniques can be used in training these machine learning methods, and to name a few, these include backpropagation, gradient descent, and least mean square gradient algorithms. Machine learning methods are categorized into two different types: supervised learning methods and unsupervised learning methods [8]. Supervised learning methods require prelabeled data for training, whereas unsupervised learning methods do not require a labeled input data for training. Both methods can be used for classification and regression tasks [8]. Machine learning is used in different applications, such as pattern recognition, self-driving cars, automation, and machine translation. There are several advantages of machine learning methods. They can detect complex nonlinear relationships between independent and non-independent variables and do not require much statistical training [10]. However, machine learning methods tend to suffer from overfitting, whereby the model performs very well only on trained data but not so well on new data [10].

The three machine learning methods that are discussed in this article are decision tree classification (DTC), artificial neural network (ANN), and deep learning (DL). DTC is a supervised learning algorithm that is used in machine learning. It consists of a tree structure that is composed of nodes and branches, and it is used for classification. ANN is a machine learning method that can be used for classification and is suitable for non-linearly separable data [15]. DL is a machine learning model that enables computers to learn by example. ANN and DL consist of neurons and weights, and the neurons are connected to each other by synapses. They also consist of activation functions, which are used to decide what neuron will be activated or deactivated, and this is specified by the architecture of the model [10]. The weight of an ANN or DL model determine how much the input affects the output.

## 3. Approach

The operation of a squirrel cage IM with broken rotor bar faults introduces asymmetry in the rotor current distribution because there will not be a current flowing through the broken rotor bar [6]. The latter initiates a disturbance in the airgap magnetic field distribution that induces voltages back in the stator winding, causing a current to flow through the winding and the supply [6,16]. The forward and backward fundamental airgap rotating magnetic fields establish line currents at frequencies of $\left(1-2s\right)f_{s}$ and $\left(1+2s\right)f_{s}$, respectively. Here, s is the slip of speed per unit, and $f_{s}$ is the frequency of the supply.

The left or lower sideband $\left(1-2s\right)f_{s}$ component is induced into the stator current by the fundamental backward-rotating field produced by the magnetic disturbance of the BRB [17,18,19,20], whereas the right or upper sideband $\left(1+2s\right)f_{s}$ results from the torque fluctuations imposed on the rotor at twice-slip frequency [16,17]. These torque fluctuations are due to cyclic variations in the current produced by the magnetic disturbance around the broken bar [16,17,21]. One cannot rely only on the lower sideband alone or the upper sideband alone to detect the BRB fault. It is a combination of both the lower and upper sidebands that should be used in the detection of BRB fault [21]. The assumption that there will not be any current flowing in the BRB is not always well founded [16]. When a rotor bar initially breaks, the resistance between the broken bar and the adjoining healthy bars is low, and as a result, the current continues to flow into the broken bar [16,17]. This current enters the BRB at the healthy side, flows along the length of the bar, and exits the BRB to the adjoining healthy bar through the laminated core, where the inter-bar resistance is low [18,19]. The work reported in [18,19] has shown that the inter-bar currents dampen the magnitude of the magnetic disturbance produced by the BRB fault, which, in turn, reduces the magnitude of the lower sideband in the current spectrum. It should also be noted that the magnitude of the lower and upper sidebands depends on loading conditions, motor slip, load inertia, and location of the BRB [20,22]. It is crucial to extract the frequency feature of the periodic signal of the current because fault-related harmonics for the BRB can be recognized in the lower and upper sidebands around the supply frequency. In this article, the frequency features of the periodical signal of the current are extracted by transforming the signal into the frequency domain using discrete Fourier Transform (DFT), which can be determined using [7].
(1)$$X\left(k\right)=\frac{1}{N}\sum_{n=0}^{N-1}x(n)e^{-j2\pi k\frac{n}{N}}$$

From Equation (1), k denotes the wave number, N denotes the samples of data, n is the integer sequence (0, 1, 2, 3…), 0≤n≤N−1, and x is the input current signature. The power density spectrum (PDS) can be obtained by transforming the stator current signal into a frequency domain signal using the DFT. This current spectrum in the frequency domain can be easily analyzed compared to the current signal represented in the time domain [7]. The generated components $f_{s}(1\pm 2s)$, which are called sidebands, are spaced around the fundamental frequency equally, with the spaces varying directly in proportion to the load [23]. The broken rotor bar frequencies can be obtained using [24].
(2)$$f_{brb}=f_{s}[k(\frac{1-s}{p})\pm s]$$
where p denotes the number of poles, although the condition of the rotor can be evaluated using the stator current. It is, however, essential to note that the frequency components $f_{s}(1\pm 2s)$ do not exist in the case of broken bars that are electrically 180 degrees from each other [25]. The sideband frequency component amplitude increases with an increase in broken bars. The Machine Learning broken rotor bar detection approach followed in this paper is illustrated in Figure 1.

The stator current waveforms of the squirrel cage induction motor operating with a broken rotor bar were obtained in the lab using a digital oscilloscope. The time domain current waveforms from the oscilloscope were transformed into the frequency domain, and the frequency features of the periodical signal were extracted by converting the signal into the frequency domain using discrete Fourier Transform (DFT). Transforming the current signal into the frequency domain was performed to analyze the sideband frequencies around the fundamental line frequency. The current spectrum data were then used to train the machine learning models.

## 4. Experimental Measurements

### 4.1. Specification and Results

The specifications and ratings of the squirrel cage IM are given in Table 1, and Figure 2 shows the healthy squirrel cage rotor and the rotors with broken bars. The experimental setup is shown in Figure 3. The setup comprises the tested three-phase squirrel cage IM coupled to a 7.5 kW asynchronous machine fed using a Siemens SINAMICS four-quadrant energy recovery AC drive, Munich, Germany. The 7.5 kW asynchronous machine serves as a mechanical load. A Tektronix MDO3000 Series four-channel digital oscilloscope, Beaverton, OR, USA, was used to store and analyze the current signatures. The start and stop operations of the three-phase squirrel cage IM were made possible by the start and stop buttons incorporated into the motor controller. The stator of the tested squirrel cage IM has a conventional double-layer (CDL) winding distributed in 36 slots to form four magnetic poles. Three motors were tested, the first with a healthy rotor, the second with 3BRB, and the third with 6BRB. The current signature obtained was then transferred to Matlab R2023a, where a C code was written to perform DFT of the motor stator current data. In the dataset used for the training models, the current signature data consists of 5000 rows, and the frequency magnitude for DFT consists of 860 rows.

### 4.2. Analysis Current Signature under Broken Rotor Bar Faults

Figure 4 shows the measured steady-state current waveforms for unloaded (load torque of 0 Nm) and loaded (load torque of 30 Nm) squirrel cage IMs. Observing the measured results in Figure 4, it is notable that for both unloaded and loaded squirrel cage IMs, the RMS values of line currents increase when the motors operate with broken rotor bars. Although the motors’ stator three-phase windings were connected in star, the third harmonics primarily occur in the Fourier components of the measured unloaded currents shown in Figure 5a.

The third harmonics of the unloaded line currents of the motor decreased when operating with broken rotor bars. The frequencies of the current components in Figure 5a,b are indicative of the influence of unbalanced rotor flux caused by the broken rotor bars, which influenced the airgap field distribution. Furthermore, the unbalanced rotor flux must be considered as a combination of positive and negative sequence rotor fluxes, rotating at a certain slip frequency, and the current harmonics can be observed as twice the slip of the frequency beside the fundamental frequency, as shown in the measured DFT zoom current spectrums shown in Figure 6.

The magnitudes of the twice-slip frequency sidebands $\left(1\pm 2s\right)f_{s}$ due to broken rotor bars are clearly shown in the measured DFT zoom current spectrums shown in Figure 6. The slip frequency of the unloaded squirrel cage induction motor is very small, and the twice-slip frequencies under the broken rotor bar operation are close to the value of the supply frequency (50 Hz). From Figure 6a, it is notable that the unloaded squirrel cage IM exhibits twice-slip sidebands at ±0.4 Hz around the supply frequency when operating either with 3BRB or 6BRB. From Figure 6b, it is clearly notable that the loaded motor exhibits twice-slip sidebands at ±1.2 Hz around the supply frequency when operating either with 3BRB or 6BRB. From Figure 6a, the sidebands are about 40 dB down on the unloaded supply current of the motor for both 3BRB and 6BRB faulty operations. In Figure 6b, the decibel difference between the sideband magnitudes and the supply frequency components for the loaded current of the motor is about 35 dB. The magnitude of the sidebands increases by 5 dB from the unloaded current to the loaded current.

## 5. Broken Rotor Bar Detection Using Machine Learning

### 5.1. Decision Tree Classification

The first node of the decision tree classification is the root node, and this is where the data start dividing into multiple sets. The other node is the decision node; it is used to make decisions and consists of numerous child nodes. The last node is the leaf node; this node does not consist of any further child nodes, and it represents a class prediction [25]. The decision tree classification (DTC) model in this study was developed using the Python programming language via the MATLAB platform. To decide the optimum splits between nodes, the model chooses the split with the highest information gain. Information gain is used to decide the assignment of decision nodes and leaf nodes. Information gain IG is calculated by subtracting the total entropy of the child nodes from the total entropy of the parent node as in Equation (3), and the entropy, which is the measure of uncertainty in the available dataset, can be computed using Equation (4).
(3)$$IG=E\left(parent\right)-\sum w_{i}E({child}_{i})$$
(4)$$E=\sum -p_{i}\;{log}_{2}\left(p_{i}\right)$$
where $p_{i}$ indicates the probability of a class, $w_{i}$ denotes the weight of the input current signature, and the subscript i indicates the class number. The minimum number of samples required to split a node in this model is two, and the maximum depth for the decision tree classifier is two. For verification and validation of the model, the input data were divided into two sets. The first set consists of 20% of available data, and it was assigned as test data. The second set consists of 80% of available data, and it was assigned as training data. Figure 7 shows the architecture of the DTC model.

### 5.2. Artificial Neural Network

The artificial neural network (ANN) model was developed using the Python programming language via the MATLAB platform. It consists of an input layer, three hidden layers, and one output layer. The model has 60 epochs with a learning rate of 0.001. The neurons are connected by synapses, and they obtain inputs from the neurons in the previous layer using Equation (5). The activation function used for the hidden layers was rectified linear unity, and it can be calculated using Equation (6).
(5)$$Z_{v}^{l}=\phi (\sum_{h=1}^{n}a_{h}w_{h}+B_{h})$$
(6)$$f\left(a\right)=max(0\;,\;a)$$
where *a* denotes the magnitude of the frequency of the DFT current spectrum, $w_{h}$ denotes the weight of the input current signature and determines how the input impacts the output of the model, $B_{h}$ denotes the bias added to the input to avoid dead cells, the subscript h represents the number of inputs, Z is the neuron, the subscript index v indicates the number of neurons, and the superscript l indicates the number of layers in the neuron network. The function returns a if the input is positive, and it returns 0 if the input is negative. In the output layer, the sigmoid activation function was used, and it is expressed as follows:(7)$$f\left(a\right)=\frac{1}{1+e^{-a}}$$

The sigmoid function normalizes the available data between 0 and 1. For verification and validation of the model, 20% of the available data was assigned as a test dataset, and 80% of the available data was assigned as a test set for validation of the ANN model. Figure 8 illustrates the architecture of the artificial neural network model used in this work.

### 5.3. Deep Learning

The deep learning (DL) model was developed using the Python language in the Tensor Flow framework. It consists of an input layer, three hidden layers, and one output layer. The model has ten epochs with a learning rate of 0.001. Just like in the ANN model, all neurons are connected by synapses, and the input to each neuron is calculated by multiplying the input with the weight, and then, adding a bias. Figure 9 shows the architecture of the deep learning model. 

The tanh activation function was applied to the first two hidden layers, and it was calculated using Equation (8). The tanh is an activation function that takes any real value and returns it in a range between −1 and 1. The leaky rectified linear unity activation function was applied to the last hidden layer, and it was obtained using Equation (9).
(8)$$f\left(x\right)=\frac{e^{a}-e^{-a}}{e^{a}+e^{-a}}$$
(9)$$f\left(a\right)=max(0.01a\;,\;a)$$

The leaky rectified linear unity function returns a when the input is positive but a very small value when the input is negative. The sigmoid function was applied to the output layer. For verification and validation of the model, 20% of the available data was assigned as a test dataset, and 80% of the available data was assigned as a test set for validation of the model.

## 6. Results and Discussion

In classification problems, the accuracy can be misleading. For example, if the data are not balanced, the model might predict the majority of one class for all cases and have a high accuracy score. Hence, for the validation of the models, a confusion matrix was used. The confusion matrix shown in Figure 10 was employed to describe the performance of the classification algorithm in machine learning models. On the other hand, Table 2 compares the BRB fault detection accuracy, precision, recall, and F1-score between the DTC, ANN, and DL models for unloaded and loaded squirrel cage IMs. In Figure 10, TP and TN represent the true positive and true negative, respectively, and FP and FN represent the false positive and false negative, respectively. 

From Table 2, it is evident that the DTC model performed well on trained data compared to the ANN and DL models when the squirrel cage IMs operated with both 0 Nm and 30 Nm load torques. Under unloaded current operation, the DTC model achieved the highest accuracy of 70% and 74% for the 3BRB fault and 6BRB fault, respectively, followed by the DL model, which reached 67% and 71% under the same faulty BRB conditions. Furthermore, under loaded current operation, the DTC model still achieved the highest accuracy of 95% and 98% for 3BRB fault and 6BRB fault, respectively. From the results depicted in Table 2, it is clearly notable that the accuracy level of all three models is significantly lower for unloaded operation when compared to loaded operation.

In [26], it was stated that under sufficient motor load circumstances, it is possible to reliably identify broken rotor bars in induction motors by examining the current signatures. However, under light load conditions, the reliability of detection is reduced. This is because under no-load conditions, the equivalent rotor resistance is high, and the slip and frequency of the rotor current are minimal, resulting in a considerable decrease in rotor magnetizing reactance; hence, the rotor current is minimal under no-load conditions. The precision of a confusion matrix was used to determine the ability of the model to classify positive values correctly. Recall was used to assess the ability of the model to predict positive values. The F1-score considers both precision and recall, and it is the harmonic mean of precision and recall.

Furthermore, in Table 2, the precision, recall, and F1-score of the DTC model are better compared with the DL and ANN models. Simply put, the results proved that the DTC model managed to achieve the highest accuracy consistently, followed by the DL model, and then, lastly, by the ANN model. Although the DTC model achieved the highest accuracy in most cases, they all were close to each other. Work reported in [9], which compared the decision tree classification and deep learning algorithms for image datasets, concluded that the decision tree algorithm outperformed the deep learning algorithm [9]. However, they stated that the data required to obtain a stable model are less for the decision tree algorithm than for the deep learning method [9]. Hence, more data could improve the accuracy of both deep learning and artificial neural networks. Also, algorithm tuning can be performed to strengthen all three machine learning methods further.

## 7. Conclusions and Future Work

### 7.1. Conclusions

This article assesses the detection of broken rotor bar faults of squirrel cage induction motors using different machine learning methods, including the decision tree classification, deep learning, and artificial neural network methods. Broken rotor bar features were detected by extracting the characteristics of the measured line-current signatures through transformation of the signal from the time domain to the frequency domain using discrete Fourier Transform. All three methods detected the magnitudes of the twice-slip frequency sidebands due to the broken rotor bar fault for unloaded and loaded squirrel cage induction motors. The decision tree classification method achieved a higher accuracy of 95% and 98% for 3BRB and 6BRB, respectively, in the detection of the magnitudes of the twice-frequency sidebands than the deep learning method, which had an accuracy of 89% for both 3BRB and 6BRB and artificial neural network method with the accuracy of 87% and 89% for 3BRB and 6BRB, respectively, under a 30 Nm load. Although the detection accuracy and precision were higher for the loaded motor than the unloaded motor, the decision tree classification method managed to also exhibit a high accuracy for the unloaded motor, with an accuracy of 70% and 74% for 3BRB and 6BRB, respectively, when compared with the deep learning method, which had an accuracy of 67% and 71% for 3BRB and 6BRB, and the artificial neural network method, which had an accuracy of 63% and 68% for 3BRB and 6BRB. The squirrel cage induction motor does not have to be heavily loaded to make an accurate assessment of the machine’s condition. There is no need to extract through measurements the characteristics of mechanical parameters, such as speed, torque, or vibration, to assess the rotor bar condition of a squirrel cage induction motor.

### 7.2. Future Work

Start-up transient and steady-state operation regimes are common in small and medium-sized squirrel cage IMs. Mostly, these small and medium-sized motors start directly online with a light or medium load. The analysis of the ability of the decision tree classification, artificial neural network, and deep learning methods to effectively detect broken rotor bar faults on small and medium-sized lightly and fully loaded squirrel cage induction motors using the characteristics of the measured line-current signature will form part of our future work. The study will consider the operation under start-up transient and steady-state regimes for direct-online small and medium-sized squirrel cage induction motors.

## Figures and Tables

**Figure 1 sensors-23-09079-f001:**
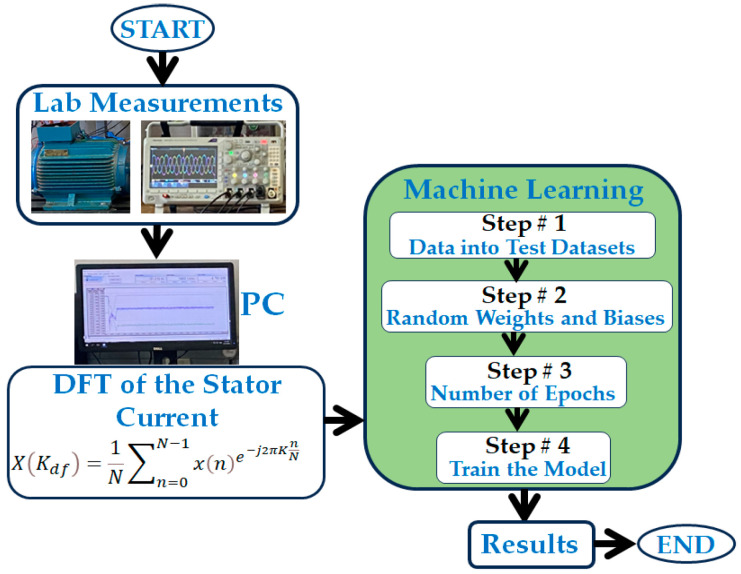
Machine learning rotor broken bar detection approach using stator current signature.

**Figure 2 sensors-23-09079-f002:**
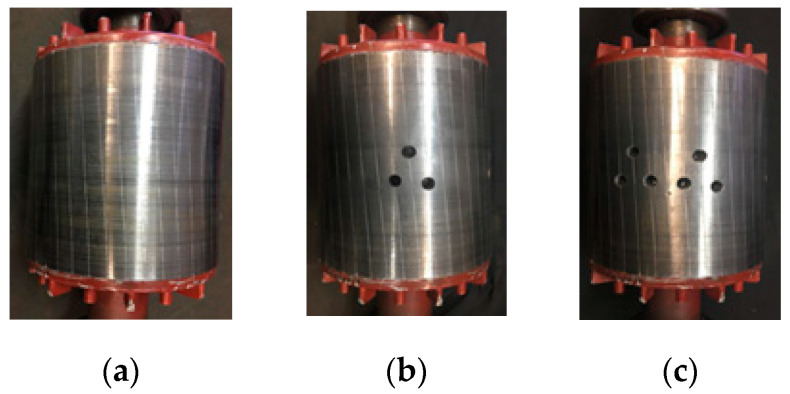
Squirrel cage rotor: (**a**) healthy, (**b**) three broken rotor bars, (**c**) six broken rotor bars.

**Figure 3 sensors-23-09079-f003:**
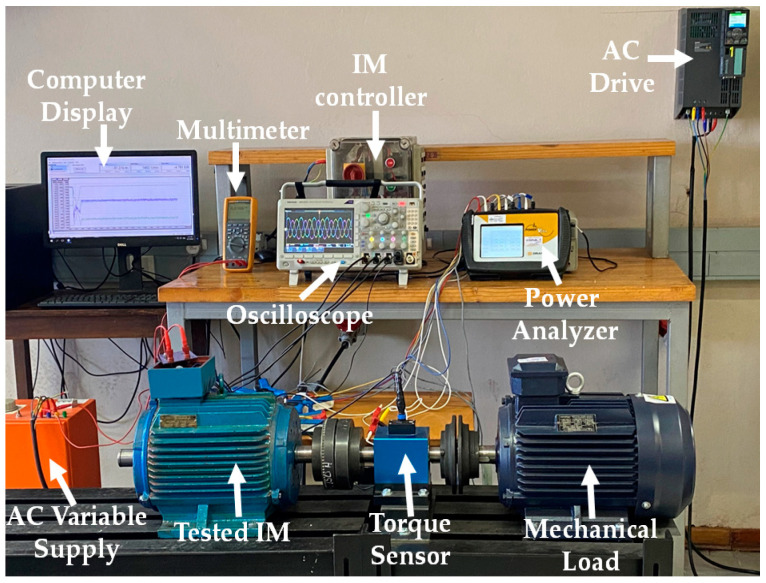
Experimental setup rig photo.

**Figure 4 sensors-23-09079-f004:**
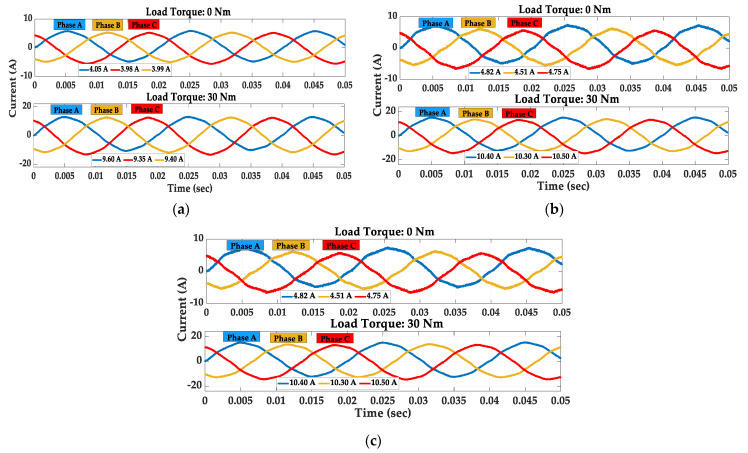
Steady-state stator currents under healthy operation: (**a**) healthy; (**b**) 3BRB, (**c**) 6BRB.

**Figure 5 sensors-23-09079-f005:**
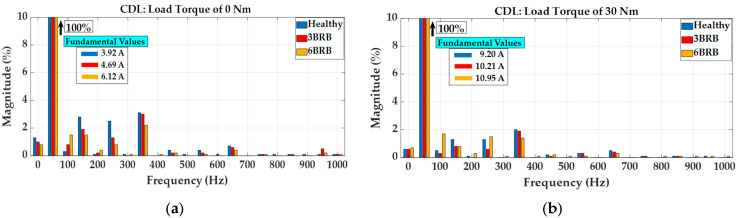
Harmonic components of phase A current profiles: (**a**) operation with load torque of 0 Nm, (**b**) operation with load torque of 30 Nm.

**Figure 6 sensors-23-09079-f006:**
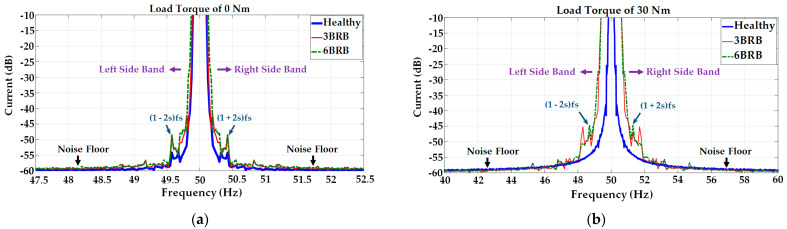
DFT zoom current spectrum of phase A under steady state of the squirrel cage IM: (**a**) operation with load torque of 0 Nm, (**b**) operation with load torque of 30 Nm.

**Figure 7 sensors-23-09079-f007:**
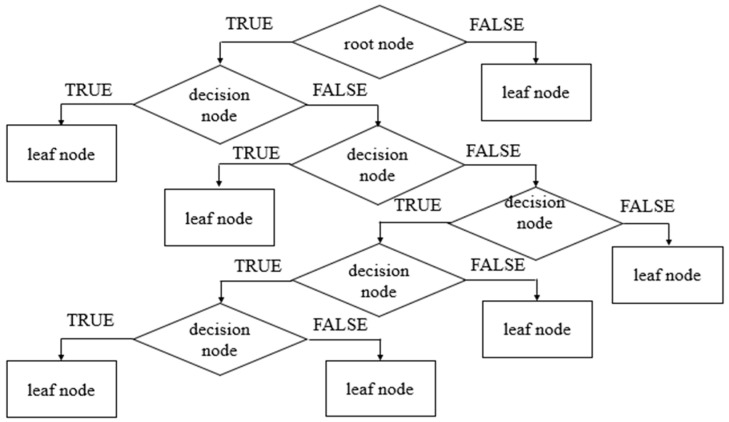
Architecture of the decision tree classification model.

**Figure 8 sensors-23-09079-f008:**
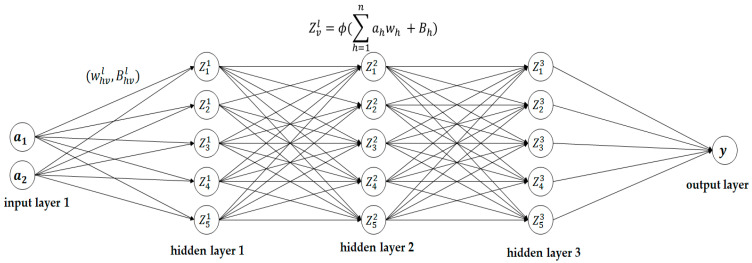
The architecture of the artificial neural network model.

**Figure 9 sensors-23-09079-f009:**
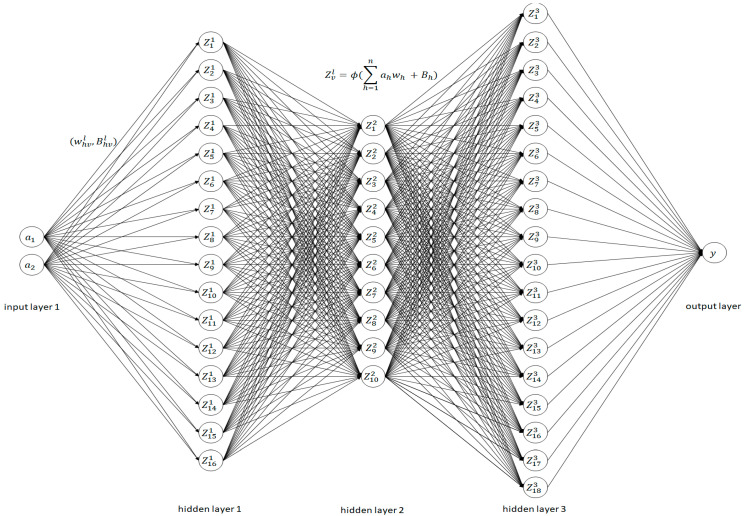
Architecture of the deep learning model.

**Figure 10 sensors-23-09079-f010:**
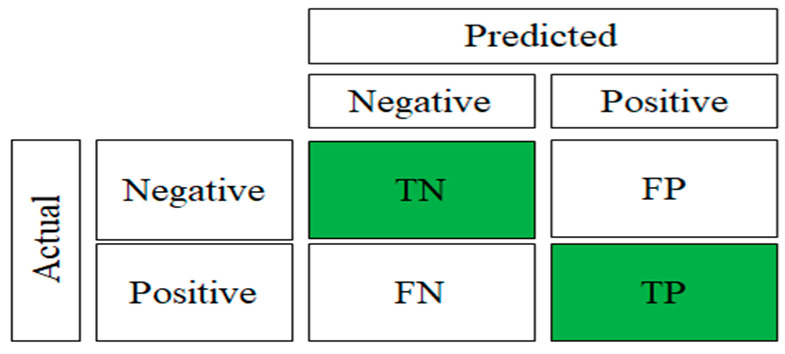
Confusion matrix.

**Table 1 sensors-23-09079-t001:** Specifications and ratings of the three-phase squirrel cage IM.

Description	Values	Unit
Rated current	12.6	A
Rated power	5.5	kW
Nominal voltage	400	V
Nominal frequency	50	Hz
Rated speed	1478	rpm
Number of pole pairs	2	-
Number of stator slots	36	-
Number of rotor bars	43	-
Number of turns per phase	54	-
External diameter	210	mm
Airgap length	0.35	mm
Core length	160	mm

**Table 2 sensors-23-09079-t002:** Comparison of BRB fault detection performance.

Performance Description	Number of BRBs	Load Torque of 0 Nm			Load Torque of 30 Nm		
		DTC	ANN	DL	DTC	ANN	DL
Accuracy (%)	3BRB	70	63	67	95	87	89
	6BRB	74	68	71	98	89	89
Precision (%)	3BRB	76	67	67	99	84	85
	6BRB	74	68	71	98	88	89
Recall (%)	3BRB	89	100	99	96	100	100
	6BRB	89	100	99	99	100	100
F1-score (%)	3BRB	82	80	78	97	91	89
	6BRB	81	81	80	99	94	89

## Data Availability

Data available on request due to restrictions eg privacy or ethical. The data presented in this study are available on request from the corresponding author.

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
