# Peer review of "Detection of Broken Rotor Bars in Cage Induction Motors Using Machine Learning Methods"

_sensors, 2023, doi:10.3390/s23229079_

Round 1

Reviewer 1 Report

Comments and Suggestions for Authors

1、How do researchers obtain and prepare training data in order to train machine learning models? What features are included in the dataset?

2、The effects of the damaged rotor bar include the asymmetric current distribution and the air gap. How do these effects lead to the sliding frequency in the harmonic? How do these sidings help with fault detection?

3、The differences in the performance of machine learning models under different load conditions are mentioned. Why is the performance poor with a light load? Does this have an impact on the status assessment of the machine?

4、It is mentioned that the decision tree classification method performs well in machine learning models. What are the advantages of this method, and why is it suitable for light-load induction motors?

5、Is it possible to further improve the performance of these machine learning methods, especially under light-load conditions, to improve the accuracy of fault detection?

Comments on the Quality of English Language

The quality of the English language is acceptable.

Author Response

I would like to thank the reviewer for the comments and recommendations. They have been very helpful as far as the strengthening of this manuscript is concerned. Herewith enclosed the responses to reviewers.

Reviewer 2 Report

Comments and Suggestions for Authors

This article uses three different machine learning methods to detect broken rotor bar in squirrel-cage asynchronous motors, which is innovative in application fields. The article has the following problems that need to be revised:

1. Please carefully review the clerical errors in the text, such as “5.1 Decision Three Classification”;

2. There are some errors with the figures in the article. The figure numbers are messed up, and there are two “Figure 6”. Figure 6 is blurry, with text misalignment. The screenshot of Figure 10 has low definition, and the aspect ratio of the text is uncoordinated.

3. The article states that there is currently no research on using DTC, ANN and DL for rotor broken bar fault detection. Are there other similar application scenarios or algorithms that can be used for comparative testing?

Comments on the Quality of English Language

Modification of English language is required

Author Response

(The authors gave the same response as above.)

Reviewer 3 Report

Comments and Suggestions for Authors

This paper evaluates the performance of three machine learning methods, namely decision tree classification (DTC), artificial neural network (ANN), and deep learning (DL), for broken rotor bar (BRB) fault detection in squirrel cage induction motors. The authors developed, applied, and studied these methods to compare their performance in detecting BRB faults.

The training data was collected through experimental measurements. The authors extracted BRB fault features from the measured line current signatures by transforming the time domain signals to the frequency domain using discrete Fourier transform (DFT).

The quality of the paper is good, methodology, procedure and results are logical and clearly presented.

The reviewer has three comments:

1. The literature review is relevant, however, some sources about MCSA - motor current signature analysis method can be added, since the authors also use current signal Fourier transforms.

2. It is recommended to add to the conclusion:

- some brief quantitative results, not only qualitative

- the applicability of the solution in terms of design features and differences of induction motors

3. It would be interesting to apply the trained models to motors using available datasets from industry (if any).

Author Response

(The authors gave the same response as above.)

Reviewer 4 Report

Comments and Suggestions for Authors

The authors presented good work in this paper. However, there are some points needs to be corrected before final acceptance. The major and minor comments are below:

Major Comments:

  1. The introduction provides a good overview of broken rotor bar faults and the need for detection methods. However, it would benefit from more clearly framing the motivation and novelty of this work compared to existing literature. What gap is this research aiming to fill?
  2. More details are needed on the data collection and experimental setup. How many motors were tested and under what operating conditions? How was fault severity varied? How was data pre-processed before training the models?
  3. The methods section should provide more details on the model architectures, hyperparameters, and training procedures for the DTC, ANN, and DL models. This will allow readers to better evaluate the results.
  4. The results comparing model performance lack statistical significance testing. Are the differences in accuracy, precision etc. statistically significant? This should be evaluated.
  5. More analysis is needed on what factors affect model performance - e.g. load condition, fault severity, data set size. How do these impact the relative performance of the models?
  6. The conclusion claims DTC works well for small, lightly loaded motors but no results are shown for this condition. This claim needs justification or removal from the conclusions.
  7. The writing could be improved for clarity and flow in some areas. For instance, the introduction jumps between intro material and describing the paper structure.
  8. References need to be formatted properly and consistently.

Minor Comments:

  1. In the abstract, briefly mention how the models were trained and evaluated.
  2. Fix capitalization inconsistencies e.g. Deep Learning vs deep learning.
  3. Table 2 is referenced before Figure 10 - this should be swapped.
  4. Expand acronyms before using e.g. ANN, DL.
  5. The conclusion should highlight the key findings, implications, and future work.

Author Response

(The authors gave the same response as above.)

Round 2

Reviewer 1 Report

Comments and Suggestions for Authors

The current version is acceptable.

Reviewer 4 Report

Comments and Suggestions for Authors

The authors have addressed all the comments successfully. Paper is ready to accept.